# Developing Lines of Queensland Fruit Flies with Different Levels of Response to a Kairomone Lure

**DOI:** 10.3390/insects13080666

**Published:** 2022-07-22

**Authors:** Maryam Yazdani

**Affiliations:** Department of Applied Biosciences, Macquarie University, Sydney, NSW 2109, Australia; maryam.yazdani@csiro.au

**Keywords:** artificial selection, *Bactrocera tryoni*, heritability, male annihilation technique, sterile insect technique

## Abstract

**Simple Summary:**

Queensland fruit fly (Q-fly) is widely recognized as one of the world’s worst economic pests of fruit. In this project, a series of artificial selection experiments were conducted to develop lines of Q-fly with different levels of response to the male-specific lure Cue-lure^®^ (CL) and to assess the heritability of this particular trait. Although lines of high and low responsive males to Cl were successfully developed through five cycles of artificial selection, the response to CL did not completely disappear. I have demonstrated that relaxing artificial selection results in the loss of 35–46% of the selection response after a further two generations.

**Abstract:**

The Queensland fruit fly (Q-fly), *Bactrocera tryoni* (Froggatt) is a serious horticultural pest in Australia because it is highly invasive and destructive. Among all pest management practices, sterile insect techniques (SIT) and male annihilation techniques (MAT) are important control options for many tephritid fruit fly pests, including Q-fly. However, simultaneous applications of MAT and SIT require the wild males to be responsive to a lure while the released sterile males remain largely unresponsive. In this study, a series of artificial selection experiments was conducted to develop lines of Q-fly with different levels of response to the male-specific lure Cue-lure^®^ (CL). After only five cycles of artificial selections, lines of high responsiveness (HR) and low responsiveness (LR) males diverging significantly in their response to the lure were developed. In the field cage experiment, the number of trapped males in fruit fly traps was significantly lower in the LR line than both the HR line and the control which supports the laboratory results. However, when artificial selection was stopped at F5 and retested after two generations, the number of unresponsive males dropped drastically compared to the rate of response of wild flies. Because the selection can be conducted only on males, it would be difficult to eliminate the dominant responsive alleles in the system without continuous selection.

## 1. Introduction

Fruit fly species (Diptera: Tephritidae) of the genus *Bactrocera* are among the most serious orchard pests worldwide. These fruit flies are found in almost all fruit-growing areas of the world, where they can cause serious damage to fruit, sometimes resulting in near-total crop failure [1]. Queensland fruit fly (Q-fly), *Bactrocera tryoni* (Froggatt) is one of the most destructive horticultural insect pests in Australia, and has expanded its distribution from its native tropical/subtropical range in coastal Queensland (QLD) and northern New South Wales (NSW) southwards into more temperate regions of NSW and Victoria (VIC) [2,3,4]. Economic losses in Australia caused by Q-fly, including crop damage, management, and ongoing surveillance, are estimated to cost AUD 300 million per year [5].

Among all pest management practices, the sterile insect technique (SIT) and male annihilation technique (MAT) are the most important control strategies for many tephritid fruit fly pests [6,7,8], including Q-fly [9]. The sterile insect technique relies on mass-production and release of sterile male insects to mate with wild females, which results in population suppression through egg infertility [10]. The male annihilation technique reduces the number of males within a population by using traps that have a male-specific lure, such as Cue-lure^®^ (CL), combined with an insecticide [11,12]. Cue-lure (4-(*p*-acetoxyphenyl)-2-butanone), a synthetic analog of the phytochemical compound raspberry ketone, is a highly attractive kairomone lure for sexually mature males [13].

The insecticidal activity of MAT devices may attract and kill the released sterile males, thereby weakening the effectiveness of SIT; therefore, MAT and SIT techniques are usually deployed in sequence. Significant improvements in both fruit fly management efficacy and cost reduction could theoretically be achieved if it were possible to employ MAT and SIT simultaneously [14]. However, simultaneous application of MAT and SIT requires wild males to be responsive to a lure while the released sterile males remain largely unresponsive.

Variations in olfactory sensitivity and responses among individuals in certain insect species have been reported at both the individual and molecular levels [14,15,16]. For example, inter-line differences in responsiveness to trimedlure, another kairomone, have been observed in Mediterranean fruit fly, *Ceratitis capitata* (Wiedemann) [17]. Furthermore, through strong artificial selection pressure, development of significant changes in olfactory responsiveness has been demonstrated in several studies. For instance, Wang, Gu, and Dorn [18] selected for the strong or weak response of a parasitoid wasp to volatiles emitted from a plant infested by phytophagous insects. Maeda and Liu [19] used a similar approach, developing lines of a predatory mite attracted to volatiles of host plants infested by a phytophagous mite species [20].

Thus, selective breeding of unresponsive males to CL is an option to suppress the response of released sterile Q-flies to CL-based MAT devices. There is support for this possibility already in studies of the oriental fruit fly, *B. dorsalis* (Hendel) [21,22]. Itô and Iwahasi [21] exposed *B. dorsalis* males to the male attractant Methyl eugenol (ME) in the laboratory and selected unresponsive males as sires. Within only two generations of such selection, they produced a line with lower ME responsiveness than the control line. Working with *B. dorsalis* in Hawaii, Shelly [22] likewise reported a consistent reduction in ME responsiveness over eight generations for several lines sired by unresponsive males. However, to date there has been no investigation of variations within or between population in male Q-fly responses to CL, and such studies now require investigation.

In this project, a series of artificial selection experiments were conducted to develop lines of Q-fly with different levels of response to CL and to assess the heritability of this particular trait. The results may open a new window to understanding the molecular mechanisms underlying the olfactory perception of CL in the future, along with the possibility of using the unresponsive line in SIT programs.

## 2. Methods

**Insects.** In November 2019, infested fruits with Q-fly larvae were collected from loquat trees (*Eriobotrya japonica*) located in Sydney, New South Wales (NSW), Australia. The infested fruits were stored on wire racks in plastic bins containing a 2-cm deep layer of fine vermiculite (Ausperl, Orica Australia (Pty. Ltd., Banksmeadow, NSW, Australia) in a controlled environment room (CER) (25  ±  0.20 °C, 65  ±  3% RH, and 11:1:11:1 light: twilight: dark: twilight photoperiod). Approximately 500 pupae of wild flies were kept in each 47.5 × 47.5 × 47.5 cm fine mesh cage and provided with a layer of liquid diet (a 60% aqueous solution of 3:1 sucrose: yeast hydrolysates) on the ceiling of the cages for the emerging adults. From ten colonies of the second generation of wild flies maintained under these laboratory conditions, flies of the same age were collected for further work. On the fourth day of adult emergence, 2000–2500 males were randomly collected from the ten colonies and reared in four separate screen cages, labelled as the four parallel cohorts A, B, C, and D. When the male flies were 14 days old, screening for responsiveness to CL was simultaneously conducted in two CERs over the three hours after dawn.

**Artificial selection of unresponsive males to Cue-lure (CL).** Twenty minutes prior to commencement of screening, a group of 30 males was transferred into a clean 47.5 × 47.5 × 47.5 cm fine mesh cage and labelled accordingly (A, B, C, or D). Meanwhile, two sets of Petri dishes containing CL were prepared for the four cages using 250 µL of 1% CL (SigmaAldrich; St. Louis, MO, USA) in paraffin oil pipetted onto 9-cm diameter filter paper and placed into the lid of a 90 mm × 15 mm Petri dish. The observations took place in two CERs simultaneously; therefore, four screening cages, each containing 30 males from colony A, B, C, or D, were placed in each CER. As Cue-lure has low volatility, a 15 cm Desk Fan (Heller Inc., Falkenberg/Elster, Germany) was used with each cage to assist with equal distribution of CL odour across each cage. Each small fan was placed 150 cm away from its allocated cage. To allow the flies to settle in the cages, the fans were turned on 2–5 min before placing the Petri dishes in the center of cages. Then, the Petri dishes with CL-trapseated paper were immediately placed facing up on the bottom of each cage in the two CERs. The flies were exposed to CL for a total of 20 min, and observations were made after 5 min and 20 min exposure by two observers who monitored the cages continuously. Flies responsive to CL were removed after 5 min, and additional flies were removed from each cage after 20 min. After 5 min exposure to CL, the flies in each cage that landed on the CL-treated filter paper were carefully covered with the Petri dish bottom and removed from the cage. In each cage, a second Petri dish cover with treated paper was then immediately set in place for another 15 min. Flies attracted to CL in the Petri dishes from each cage were counted and then transferred to a separate cage in order to establish four colonies of the highly responsive (HR) line for each replicate (F0: HR A, HR B, HR C, and HR D). The second set of Petri dishes containing CL were kept in the cages for 15 min; then, males attracted to the CL from minutes 5 to 20 were removed from the cages and discarded. The remaining males in each cage, which were ostensibly unresponsive to CL, were transferred to a holding cage, supplied with food and water, and kept for rescreening. Two days after the first screening, rescreening was conducted on males unresponsive to CL alone by following the same procedure; however, for the rescreening, all males attracted to CL were discarded. The remaining unresponsive males in each screening cage were counted and transferred to a clean cage in order to establish the low responsiveness (LR) line for each replicate (F0: LR A, LR B, LR C, and LR D). Therefore, each of the four males that failed to respond to CL in the double-exposure experiment were used as low responsiveness sires, while males that responded to CL during the initial 5 min of exposure were used as high response sires for the next generation (F1). The selected males (LR or HR) belonging to each replicate line of flies in two CERs were pooled and mated with an equal number of females emerging from the same colony as that from which the screened males originated. When the Q-fly adults were 21 days old, eggs were collected from each replicate colony using oviposition devices [23] (250-mL low-density polyethylene (LDPE) plastic bottles that had numerous puncture holes for oviposition and contained ca. 10 mL of tap water). Eggs were washed from the oviposition device and then poured into a 50-mL plastic tube. Then, 250 µL of eggs collected from each colony were seeded separately onto 150 mL of a recently-prepared, gel-based larval diet [24] in 500 mL clear plastic rearing trays (17.5 × 12 × 4 cm) to establish the offspring of the first selection (F1).

This selection regime was repeated for another five consecutive cycles to develop high and low responsive lines and to test the rate of wild fly response to CL. However, due to a heat shock caused by technical failures in the CERs where the colonies were being reared, high mortality rates occurred in the colonies, and therefore rescreening was not conducted for the third cycle of selection (F3). To test the stability of the response patterns in low and high responsive lines over generations, as described above, selection was stopped for two generations (F6 and F7), then the responsiveness was retested. For this purpose, only males from both high and low lines of replicates C and D (from both CERs), which showed a constant pattern of response to CL over four consecutive cycle selections, were subjected to this bioassay. Therefore, about 200 males from LR C, LR D, HR C, and HR D colonies were subjected to the screening process in eight replications. However due to the high mortality in HR C males for unknown reasons, the data for HR C were not included in the analysis; thus, only HR D, LR C, and LR D were tested.

The low and high responsiveness of male Q-flies to CL was calculated using the following equations:$$\mathrm{LR}\;\mathrm{to}\;\mathrm{CL}\;(\%)=\frac{\;\mathrm{Number}\;\mathrm{of}\;\mathrm{male}\;\mathrm{flies}\;\mathrm{not}\;\mathrm{trapped}\;\mathrm{by}\;\mathrm{CL}\;\mathrm{after}\;20\;\mathrm{mins}\;\mathrm{exposure}}{\mathrm{Number}\;\mathrm{of}\;\mathrm{test}\;\mathrm{male}\;\mathrm{flies}}\;\times \;100$$
$$\mathrm{HR}\;\mathrm{to}\;\mathrm{CL}\;(\%)=\frac{\;\mathrm{Number}\;\mathrm{of}\;\mathrm{male}\;\mathrm{flies}\;\mathrm{attracted}\;\mathrm{to}\;\mathrm{CL}\;\mathrm{during}\;\mathrm{initial}\;5\;\mathrm{mins}\;\mathrm{of}\;\mathrm{exposure}}{\mathrm{Number}\;\mathrm{of}\;\mathrm{test}\;\mathrm{male}\;\mathrm{flies}}\;\times \;100$$

Statistical analyses were conducted to compare the rate of non-responsiveness/responsiveness to CL in the lines among the generations using one-way ANOVA, followed by post hoc mean comparisons using LSD tests. A non-parametric Levene’s test was used to verify the equality of variances in the samples. Where the data did not meet the assumptions of a parametric distribution, an independent sample Mann–Whitney U test was carried out. All analyses were conducted using IBM SPSS statistics version 28.

**Large field cage experiment.** A field trial was set up using offspring of the fifth selection cycle (F5) of selected flies from colonies C and D. High and low responsive males from the C and D lines as well as flies with parents that were unselected (i.e., control) were used. Two days prior to adult eclosion, pupae from each line and from the control were dyed with five distinct fluorescent dyes, Comet Blue 60, Astral pink 1, Lunar Yellow 27, Stellar Green 8, and Blaze 5 from (Swada, Chelshire, UK), at a rate of 1 g of dye per 100 g of pupae in order to identify the different lines and colonies. Dyed flies accumulate each fluorescent dye in their ptilinum during eclosion [25]. The dye can be seen via microscopic examination under UV light [26]. The colours were rotated among treatments between replicates. A group of 100 males from each line and the control were separated from the females well before reaching sexual maturity at 5–7 days of age. Therefore, five groups of 100 dyed flies were kept in five holding cages and supplied with food and water for each replication. The field work was conducted in large enclosures located at the Macquarie University Campus, Sydney, NSW, Australia, (33°46′6.92″ S 151° 6′48.78″ E) during November 2020, when daily minimum and maximum temperatures ranged from 16 to 25 °C. Each enclosure consisted of metal frames 8 m wide, 24 m long, and 5 m high covered in white mesh. The enclosure contained six fruit trees, such as lemon and mulberry, at various growth stages. Six Fruit Fly Trap Protraps^®^ (BUGS FOR BUGS, Toowoomba, Queensland, 4350 Australia) were suspended 8 m apart in the field cage. Traps were supplied impregnated with CL and malathion (Maldison^®^) as insecticide. Then, 100 males from each LR, HR (C and D), and the control were randomly collected; in total, 500 mixed flies were released into the large field cages. After releasing the flies into the field cages, the dead flies in each holding cage were collected and sorted based on their colour under UV light and the number of dead flies belonging to each line was recorded. To sustain the released flies during the field cage experiment, sponges extending through the lids of 1 L plastic containers containing 15% aqueous sucrose solution were suspended from the ceilings of the field cages. Every 24 h, the traps were emptied, collected flies were sorted based on their colour under UV light, and the number of captured flies belonging to each line was recorded. There were a total of eight collection days for each field cage experiment. The field cage experiment was repeated six times. The mean proportions of trapped males to released flies were compared between the LR and HR lines and the control using one-way ANOVA and, where appropriate, LSD mean separation tests. All tests were carried out in IBM SPSS statistics version 28.

## 3. Results

The response pattern to CL from F0 to F5 was not similar across all four replicates; only in replicates C and D were the response patterns of both HR and LR lines similar (Figure 1 and Figure 2). When the first cycle of selection commenced, about 40% of the F0 males responded to CL within the initial five minutes of exposure. However, after the fifth cycle of artificial selection (F5) this proportion increased to 70–80% in replicates C and D of the HR lines without changing drastically in either of the LR line replicates, remaining about 40% across all selection cycles (Figure 1). The mean percentage of responders in HR lines during the initial five minutes of exposure to CL increased over the generations, and was significantly higher in F5 than F0 (*df* = 5; *F_A_* = 18.8_,_
*F_B_* = 37.3, *F_C_* = 25.8, *F_D_* = 35.3; *p* = 0.0001), although the pattern of response did not remain steady in replicates A and B (Figure 1). Inexplicably, in the LR lines the response to CL in the initial five minutes of exposure showed an increasing trend over the selection cycles in replicates A and B (*df* = 5; *F_A_* = 14.3, *F_B_* = 10.6; *p* = 0.0001). However, as expected, the mean proportion of responsive males from LR lines in replicates C and D did not increase over F0 to F5 (*df* = 5; *F_C_* = 5.0, *F_D_* = 2.2; *p* = 0.06) (Figure 1).

The frequencies of unresponsive males to CL in both HR and LR selected lines were compared across generations. The results showed that after the second cycle of artificial selection the proportion of males unresponsive (F2) to CL in LR was significantly greater than for HR in replicates C and D (*df* = 28, *F_C_* = 23.68, *p* = 0.0001; *F_D_* = 12.55, *p* = 0.001), and the pattern remained constant during the next four selection cycles. The proportion of unresponsive males in replicate C of the LR line increased after the second round of artificial selection and remained consistently high. In contrast, the proportion of unresponsive males in the HR lines dropped in replicates C and D, to 13 and 10% respectively, and stayed statically lower than for the LR lines (*df* = 34, *F_df,C_* = 56.85, *p* = 0.0001; *F_df,D_* = 91.31, *p* = 0.0001).

The proportion of males that did not land on the Petri dish containing CL during two exposure periods (20 min) increased from 25% in wild flies to 65% and 54% in LR lines for replicates C and D, respectively (Figure 2). This proportion remained consistently high across the fourth and fifth screenings (F4 and F5), while staying constant at 25–30% for replicates A and B across all generations. After stopping the artificial selection for two generations and then recommencing the screening procedure, the percentage of unresponsive males for both LR replicates C and D dropped to about 20% and remained parallel with the wild flies (Figure 2).

The total number of responders from all four (A, B, C, and D) colonies for each selection cycle were pooled (Figure 3). After performing the second artificial selection, the per-centage of total unresponsive males was increased in the LR lines and found to be significantly higher than unresponsive males in the HR lines (F2: *df* = 119; *F* = 24.53; *p* = 0.001, F4: U (NHR = 56, NLR = 96) = 4425.5, Z = 6.64, *p* < 0.001; F5: U (NHR = 48, NLR = 96) = 4238, Z = 8.2, *p* < 0.001). When the number of unresponsive males in the four colonies (A, B, C, and D) were pooled, the mean percentage of unresponsive males across the LR and HR lines in the final selection cycle (F5) was respectively higher and lower than for the wild Q-fly (*df* = 110; *F* = 13.62; *p* < 0.001, *df* = 194; *F* = 7.54; *p* = 0.007).

The results from the field-cage experiment indicated that the numbers of trapped males responsive to CL at 24 h and eight days after release were significantly different between the LR and HR lines (*df* = 2; F24 h = 2.84; *p* = 0.0001; F8 days = 14.44, *p* = 0.001). Males in the HR line responded to CL more than the control on the first day after release, and were higher in number than the total number of trapped flies across the eight collection days, although the trapped numbers did not show statistical differences in comparison with the control (*df* = 16, F24 h = 2.83; *p* = 0.112; F8 days = 2.09; *p* = 0.167). The number of responders in LR was significantly lower than the control on day one (*df* = 16; *F* = 9.26, *p* = 0.008). In addition, the average total number of trapped flies during the eight collection days was significantly lower in the LR line than in the control (*df* = 16, *F* = 5.32, *p* = 0.03) (Figure 4).

## 4. Discussion

Artificial selection can be a powerful method, allowing us to examine divergence in odour-guided behavioural responses and determine how selection for attraction may result in responses correlated with other traits [27]. Here, we conducted artificial selection experiments to independently select for high and low attraction of Q-fly males to CL. High and low responding lines were developed after the second cycle of artificial selection. The results here are additional evidence that not all individuals of a species have the same response to lures. Similarly, Wang et al. [18] developed two lines of the braconid wasp, which after only a single generation produced lines diverging significantly in their response to a hexane extract from herbivore-infested plants. Itô and Iwahashi [21] suggested that persistence of populations of invasive *B. dorsalis* in Japan despite prolonged male annihilation was due to selection for ME-insensitive males. In addition, artificial selection for breeding of ME-insensitive males [22] has provided further evidence to support the results presented here.

The proportion of selected males unresponsive to a double exposure of CL increased from 24% in the original colony to 54% and 65% (F5), respectively, in two of the four replicate lines, C and D, in successive generations. These data agree with previous findings by Shelly [22], Guo et al. [28], and Liu et al. [29], who showed that the responsiveness of male *B. dorsalis* to ME could be reduced via artificial selection under laboratory conditions in 6–12 generations. In this study, the proportion of unresponsive Q-fly males to CL increased to 54–65% between the second and fifth artificial selection cycles, while for *B. dorsalis* [22], in two of the four test lines the proportion of unresponsive males increased from approximately 5% to 25% in twelve generations. In all previous studies, artificial selection attempted to develop only unresponsive lines; however, in my study, both LR and HR lines were developed and compared under both laboratory and field conditions. The greatest proportion of unresponsive males to CL stabilised at about 65% for LR lines and 13% for HR lines. The highest proportion of unresponsive Q-fly males to CL in this study was twice that of the highest proportion of unresponsive males that Shelly [22] reported for *B. jarvisi* [22]. The field cage data showed a similar response rate, which supports the laboratory bioassay results (Figure 4).

Hager [30] proposed that behaviour may evolve over many generations when there is behavioural variation among individuals of a population, assuming variation and heritability rates are high enough. Shelly [22] and Guo et al. [28] reported that the proportion of males with no response to ME could be increased persistently across several cycles of artificial selection under laboratory conditions, as the characteristics of defective individual olfactory sensory physiology to ME may be passed on to offspring. Likewise, the results here show variation in chemosensory-related behaviours in the Q-fly population in response to CL, a variation that is heritable (Figure 1 and Figure 3).

Shelly’s study [22] of selection for non-responsiveness of oriental fruit flies to ME suggests that such non-responsiveness occurred in only two of four replicates, and the control and selected males showed consistent differences in responsiveness to ME. Similar results were observed during this experiment, where selected males in both the low and high responsiveness lines of two of four replicates did not show consistent patterns of responsiveness to CL, while the selected males from the other two replicates (C and D) responded significantly differently. All four replicates originated from the same source of wild flies. Why such differential selection might occur in certain select lines and not others remains unclear. However, when the collected data from all four replicates of Q-fly were pooled the significant differences in responsiveness to CL were still observed and were even more pronounced between two HR and LR lines (Figure 3 and Figure 4).

In Shelly’s study [22], the response to selection was different in the field than in the laboratory. They assumed that the selection protocol effectively inhibited the mechanisms associated with close-range attraction in *B. jarvisi* without concurrently affecting factors involved with long-range attraction. Shelly [22] concluded that it is possible that “responsiveness” to ME is a composite trait involving variable thresholds for physiological and/or behavioural responses with varying distances and concentrations to the lure. In contrast with their observation, in this study, the results for the long-range attraction in the field cage corresponded with the close-range attraction in the screening cages under laboratory conditions (Figure 1 and Figure 4).

The selection experiment did not result in the complete disappearance of the male flies’ attraction to CL, which agrees with the results of Shelly’s [22] selection experiments, along with those of Guo et al. [28] and Liu et al. [29]. However, the new results demonstrate that artificial selection drastically decreased the number of Q-fly responders to CL (Figure 2). Due to the lack of resources, the selection cycles could not be run for additional generations to determine whether complete disappearance of the male flies’ attraction to CL would occur.

The attraction of adult male Q-flies to raspberry ketone (RK), which is emitted by certain plants, is related to sexual performance; males exposed to RK can gain substantial mating advantages [31]. Cue-lure, an analog of RK, is spontaneously converted to RK in the presence of moisture [32], and is attractive to mature males of certain *Bactrocera* and *Zeugodacus* fruit flies [32,33]. The robust attractiveness of male attractants seems to be intimately associated with male sex pheromone production and accumulation [34,35]. In fact, phytochemical lures are associated with sexual selection governed by both female preference and male competitive mechanisms [34]. When females were allowed to assess male attractiveness and choose appropriate mates, divergence in the population did not increase [36]. Hein et al. [37] observed that a multivariate male trait preferred by *D. serrata* females can respond to selection and result in the maintenance of male mating success [37]. As Q-fly females are not responsive to CL, they were not subjected to selection in my study. However, the progeny can inherit the gene variant responsible for the responsiveness to the attractant from both sexes. These facts could explain why complete disappearance of the male attraction to CL did not occur in my study. It is likely that even if the selection cycles were continued for more rounds the complete disappearance of the male flies’ attraction to CL would not occur, as females will eventually assess the males for mating.

The results support this assumption; when artificial selection was stopped at F5 and retested after two generations, the number of unresponsive males dropped drastically compared to the rate of response of wild flies in their second generation, and did not differ significantly from the HR line (Figure 2). Hein et al. [37] have demonstrated that relaxing artificial selection of *D. serrata* results in the loss of 52% of the selection response after a further five generations, demonstrating that the response under artificial sexual selection was opposed by antagonistic natural selection.

Liu et al. [29] showed that the responsiveness to ME of male *B. dorsalis* flies is controlled by genes associated with olfactory processing; by silencing odorant-binding protein, the mature males’ responsiveness to ME was reduced This phenomenon needs to be investigated for Q-fly as well, because the responsiveness of male Dacini flies to their respective attractants are different across species [38]. However, we need to examine whether selecting for males that are unresponsive to kairomone would compromise their foraging abilities and sexual behaviors. A comparative study of the chemosensory receptors responsible for male attractants should provide insights into the structure–activity relationships from the point of view of the structural properties of chemosensory receptors [35]. The olfactory response to CL might be controlled by several genes, with non-responsiveness to CL likely to be a recessive trait at the assay concentration used, while selection can only be undertaken on males. It would be difficult to eliminate the dominant responsive alleles in the system because we cannot screen the females, and without continuous selection the flies would eventually return to their wildtype state (Figure 2).

This is the first investigation of within-population variations in male Q-fly responses to CL. Samples from six generations of selected flies from each colony and a control group have been collected for future research at molecular level. Therefore, based on the data available in this study along with the collected samples, future research might profitably examine the molecular mechanism and transport pathway underpinning how CL is able to lure Q-flies. From the selection responses observed in this study, it can be assumed that the alleles conferring response to CL in this Q-fly population are segregated, although two lines of evidence suggest that the alleles may instead segregate at low frequency in the base population. By conducting genome-wide association analysis and transcriptome comparison of baseline expression levels of genes between the LR, HR, and control lines, it will be possible to identify the genes regulating the production of the odorant-binding proteins responsible for the detection of CL and related compounds in Q-flies (Liu et al. [29]).

Developing a deeper understanding about the genetics of the olfactory system in *B. tryoni* would pave the way for more detailed studies of ways to improve the quality of SIT and MAT techniques in the future. If the genetic makeup influenced by the lure could be identified, this would make it theoretically possible to manipulate the genome of an SIT-selected male line such that it expresses the benefits of lure exposure without the need for actual exposure. Available RNA interference (RNAi) and CRISPR technologies are powerful tools to deactivate genes regulating the production of the odorant-binding proteins responsible for the detection of CL by Q-fly males [39,40,41,42].

## Figures and Tables

**Figure 1 insects-13-00666-f001:**
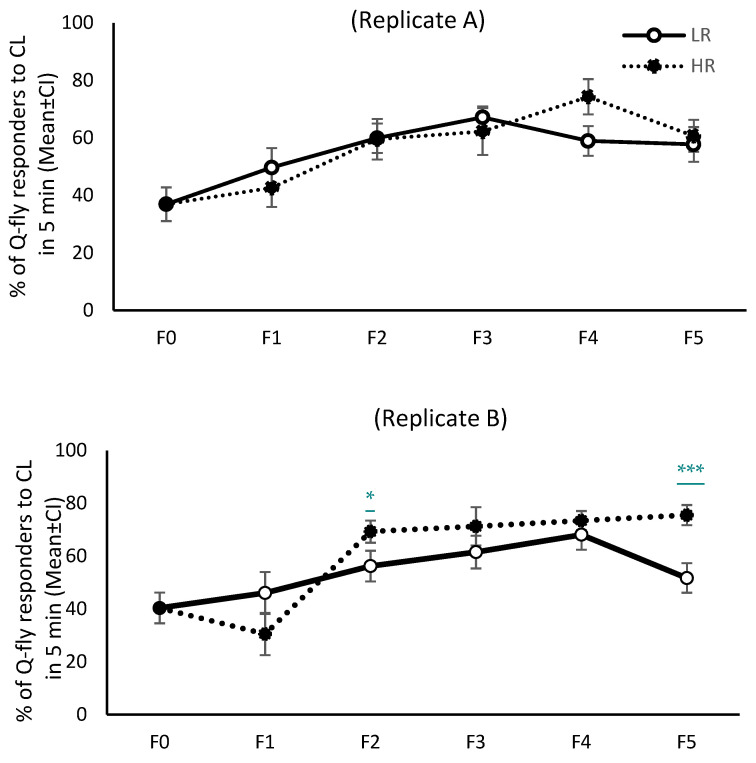
Mean percentage of Q-fly male responders to Cue-lure during the initial 5 min of exposure in four cohort replicates, (**A**–**D**). The solid line depicts the low responsive line (LR) and the dotted line shows the high responsive line (HR). Error bars represent 95% confidence intervals. Symbols *, **, and *** indicate the level of statistical significance between the mean percentage of LR and HR responders to CL at *p* < 0.01, 0.001, and 0.0001 respectively.

**Figure 2 insects-13-00666-f002:**
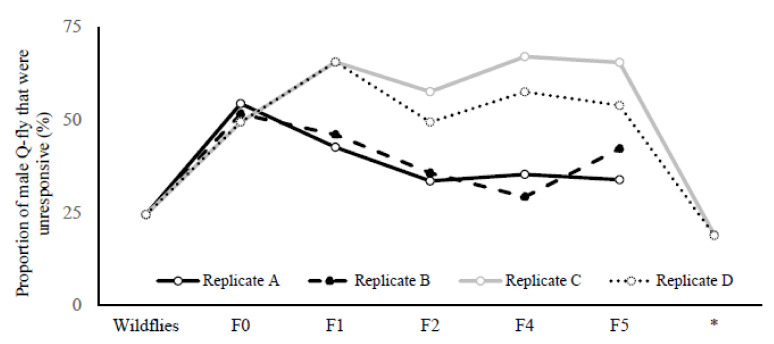
Proportion of males among 600–900 males for each replicate that did not land on the Petri dish containing Cue-lure during two exposure periods (20 min) in a controlled environment room (CER) spaced two days apart, plotted against selection cycles from the onset of the experiment. * indicates the results from retesting the selected groups after stopping the selection for two cycles.

**Figure 3 insects-13-00666-f003:**
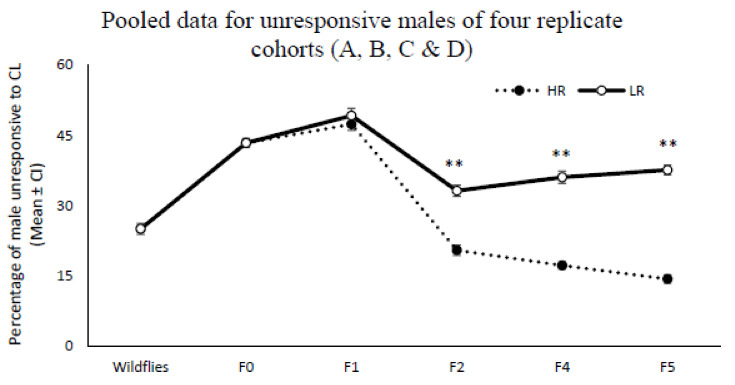
Mean percentage of pooled data for unresponsive males from four replicate cohorts (A, B, C, and D) for low responsive lines (LR) and high responsive lines (HR). Total number of Q-fly males subjected to screening in each selection cycle: Wild flies = 1100; F0 = 2955; F1 = (HR, 1892; LR, 2145); F2 = (HR, 1412; LR, 2127); F4 = (HR, 1652; LR, 2715); F5 = (HR, 1423; LR, 2748). ** indicates statistical significance between average unresponsive males from LR and HR lines at *p* < 0.01.

**Figure 4 insects-13-00666-f004:**
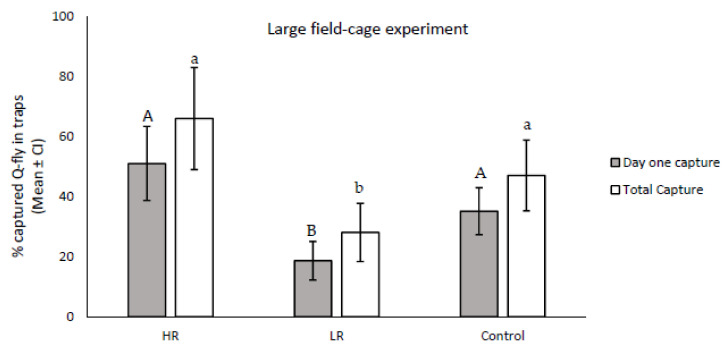
Mean proportion of trapped Q-flies from the low responsive line (LR), high responsive line (HR), and the control 24 h after release (gray bars) and after eight collection days (white bars) in the field cage. The error bars represent 95% confidence intervals. Lowercase and uppercase letters indicate significant differences between white bars and gray bars, respectively. Numbers followed by different letters are statistically different; *p* ≤ 0.05 (Tukey’s test).

## Data Availability

The data presented in this study are available on request from the corresponding author Maryam Yazdani, upon reasonable request.

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
