# Peer review of "Developing Lines of Queensland Fruit Flies with Different Levels of Response to a Kairomone Lure"

_insects, 2022, doi:10.3390/insects13080666_

Round 1

Reviewer 1 Report

This paper is well written so I will only mention the major aspects that need attention. Minor corrections are given throughout the annotated manuscript.

Recently, Q-fly moved further south and if I recall correctly, even reached Tasmania. I think this should be mentioned in the introduction as it increases the significance of the pest.

The author suggests that non-responsive flies to CL could be used in SIT at the same time that MAT is used to kill responsive wild flies.  However, in some SIT applications the over-flooding ratio is measured by use of traps so sterile males and wild males need to be equally attracted to the lure used.  If an alternative lure to CL could be used that non-responsive CL flies would be attracted to this could avoid this difficulty.

The figures in the manuscript are mixed up and Figure 3 is duplicated. Figure 1 should have 4 small graphs but it has an additional large graph at the bottom.  This large graph in Figure 1 is actually Figure 2 and needs to be moved down to where the first Figure 3 appears with the caption for Figure 2. This first Figure 3 should be deleted because it is repeated below with the correct Figure 3 caption. 

Some details need to be added to the Methods to clarify what was done, particularly with regard to replicates.

The term field-cage is sometimes hyphenated and sometimes not.  This needs to be consistent.

The reference citing style is also inconsistent. Sometimes all the authors are mentioned and other times et al. is used. It will be easier to use et al. throughout. 

It is possbile that in selecting for males that are non-responsive to CL some other abilities related to searching for mates may be compromised.  This may therefore have a negative effect on the use of SIT if released males are less able to find wild females.  This could be mentioned in the discussion.

Author Response

Dear Editors,

Thank you for the opportunity to revise our manuscript (Manuscript ID: insects-1760419), entitled “Developing lines of Queensland fruit flies with different levels of response to a Kairomone Lure”.

I appreciate the careful review and constructive suggestions. The simple summary added as requested. I believe that the manuscript is substantially improved after making the suggested edits. Following this letter are the editor and reviewer comments with my responses in italics, including how and where the text was modified (PDF file). Changes made in the manuscript are marked using track changes.

Please address all correspondence concerning this manuscript to me at maryam.yazdani@csiro.au .

Thank you for your time and consideration.

Sincerely,

Maryam Yazdani

Reviewer #1 Comments and Suggestions for Authors

This paper is well written so I will only mention the major aspects that need attention. Minor corrections are given throughout the annotated manuscript.

  • All the minor corrections have been addressed throughout the annotated manuscript. The reply to the reviewer’s comments added as well.

Recently, Q-fly moved further south and if I recall correctly, even reached Tasmania. I think this should be mentioned in the introduction as it increases the significance of the pest.

  • The incursion happed in Tasmania but as of 30 March 2019, the whole of Tasmania is once again fruit fly free. Please check the Department of Natural Resources and Environment Tasmania website

The author suggests that non-responsive flies to CL could be used in SIT at the same time that MAT is used to kill responsive wild flies.  However, in some SIT applications the over-flooding ratio is measured by use of traps so sterile males and wild males need to be equally attracted to the lure used.  If an alternative lure to CL could be used that non-responsive CL flies would be attracted to this could avoid this difficulty.

  • As far as I know there is no alternative attractant for MAT. Akter et al. 2017 (https://doi.org/10.1371/journal.pone.0184086) pointed some alternatives ways to CL trap for monitoring of released sterile flies. I didn’t add the information addressing the reviewer comments here, as the information related to monitoring SIT is not directly related to my results but happy to add few sentences if the reviewer would recommend

The figures in the manuscript are mixed up and Figure 3 is duplicated. Figure 1 should have 4 small graphs but it has an additional large graph at the bottom.  This large graph in Figure 1 is actually Figure 2 and needs to be moved down to where the first Figure 3 appears with the caption for Figure 2. This first Figure 3 should be deleted because it is repeated below with the correct Figure 3 caption. 

  • The figure captions were not in the right orders, this has been fixed in the revised version

Some details need to be added to the Methods to clarify what was done, particularly with regard to replicates.

  • More detailed information has been added (marked using track changes) to the methods to clarify the raised issues

The term field-cage is sometimes hyphenated and sometimes not.  This needs to be consistent.

  • The term field-cage is hyphenated across the text

The reference citing style is also inconsistent. Sometimes all the authors are mentioned and other times et al. is used. It will be easier to use et al. throughout. 

  • I used et al., only when the number of authors were less than 4. However, in the final version the references would be appear in number

It is possbile that in selecting for males that are non-responsive to CL some other abilities related to searching for mates may be compromised.  This may therefore have a negative effect on the use of SIT if released males are less able to find wild females.  This could be mentioned in the discussion.

 The required information has been added in the discussion as below:

Liu et al., (2017) showed the responsiveness to ME of male B.dorsalis flies is controlled by genes associated with olfactory processing, and by silencing odorant-binding protein the mature males’ responsiveness to ME was reduced This phenomenon needs to be investigated for Q-fly as well, because the responsiveness of male Dacini flies to their respective attractants are different across species (Tan, Nishida, Jang & Shelly, 2014). However, we need to examine whether selecting for males that are unresponsive to kairomone, would compromise their foraging abilities and their sexual behaviors.

Reviewer 2 Report

The article is interesting and well written and well discussed in the context of literature. However, in my opinion, it suffers from certain limitations that should be addressed before publication. The proposed research study was not successful in selecting a population of Q-fly suitable for the proposed goal. Also, it points out that this objective could be hardly achievable due to the high variability of the response to the stimulus and due to the fact that heritability seems to be limited. The fact that selection can be conducted only on males but that mating with unselected females may lead to lose the effects of the selection is a key constraint. Therefore, conclusions seem too optimistic as they are not sufficiently supported by data. In addition, the unlucky events that have limited the number of repetitions available for the statistical analysis reduce the possibility to generalize the discussed mechanism of selection and do not assure to be certain about repeatability. Regarding methods, a larger and ventilated environmental context could have increased the value of the selection process by limiting the effects of randomness. My advise is to repeat the tests to increase the dataset and to improve the experimental set up, and then, to revise the discussion to better take into account the obtained results that can be discussed under a biological point of view as well as in the context of practical perspectives.

There are sentences in the discussion that are too speculative, as an example, those starting at line 17 of page 10.

Also conclusions are not self sufficient and are too much speculative. They should highlight sounding findings and not hypothesize mechanisms of selection that have not been directly studied in the manuscript.

Author Response

Dear Editors,

Thank you for the opportunity to revise our manuscript (Manuscript ID: insects-1760419), entitled “Developing lines of Queensland fruit flies with different levels of response to a Kairomone Lure”.

I appreciate the careful review and constructive suggestions. The simple summary added as requested. I believe that the manuscript is substantially improved after making the suggested edits. Following this letter are the editor and reviewer comments with my responses in italics. Changes made in the manuscript are marked using track changes.

Please address all correspondence concerning this manuscript to me at maryam.yazdani@csiro.au .

Thank you for your time and consideration.

Sincerely,

Maryam Yazdani

  • After only five cycles of artificial selections, lines of high (HR) and low responsiveness (LR) males diverging significantly in their response to CL (F5: n= 144; P<0.001), so in this study low and high responsive lines were successfully developed. These data agree with previous findings by Shelly (1997), Guo et al. (2010), and Liu et al. (2017), who showed that the responsiveness of male B. dorsalis to ME could be reduced via artificial selection under laboratory conditions in 6–12 generations. However, they didn’t test if after relaxing the selection cycle the level of response would stay constantly low or would increase as we tested and observed.

  • This is the first investigation of within population variations in male Q-fly responses to CL. In the discussion we just suggested that the collected samples from six generations of selected flies from each colony and a control group would be useful for future research at molecular level. Therefore, based on the data available in this study and the collected samples, future research might profitably examine the molecular mechanism and transport pathway underpinning how CL is able to lure Q-flies. Similar approached was conducted by Liu et al., (2017) for B.dorsalis.

  • Shelly 1979 used the larger environment for their selection experiments, so I assumed that more controlled environment would give a better result. In my study the proportion of unresponsive Q-fly males male Q-fly non-responders to CL increased to 54%-65% between the second and fifth artificial selection cycles, while for B. dorsalis (Shelly, 1997), in two of the four test lines, the proportion of unresponsive males non-responding males increased from approximately 5% to 25% in 12 generations in the field cages.

  • It is impossible to repeat this work, completing this experiment took about 17 months, and the number of Q-fly males subjected to screening in each selection cycle were much higher than similar studies: Wild flies=1100; F0=2955; F1=(HR,1892; LR,2145); F2=(HR,1412; LR,2127); F4=(HR,1652; LR,2715); F5=(HR,1423; LR,2748).

  • Not sure which statement the reviewer referred to (page 10 line 17: The attraction of adult male Q-fly to raspberry ketone (RK), emitted by some plants, is related to sexual performance since males exposed to RK can gain substantial mating advantages (Shelly 2010).)

  • I deleted the conclusion (not required for this journal)

Round 2

Reviewer 2 Report

The article has been improved and it only requires minor revisions. Certain statements still appear too speculative but they can be easily revised.

In Simple Summary:

“So, we assume unresponsiveness to CL is likely to be a recessive trait” is an hypothesis but there are no data in the article to support it. Therefore, I suggest to remove the sentence from this summary

In Responses to the reviewers the author writes: “Shelly 1979 used the larger environment for their selection experiments, so I assumed that more controlled environment would give a better result. In my study the proportion of unresponsive Q-fly males male Q-fly non-responders to CL increased to 54%-65% between the second and fifth artificial selection cycles, while for B. dorsalis (Shelly, 1997), in two of the four test lines, the proportion of unresponsive males non-responding males increased from approximately 5% to 25% in 12 generations in the field cages”.

The studies are not comparable because different species and different attractant are involved. Reported results are not sufficient to generalize the mechanism of selection. Maybe, my comment was not clear but, what I meant was that, in a small environment, it is easier for an unresponsive male to land on the attractant even if not, or just less, attracted. Also, small environments have a higher possibility to be saturated by an attractant that is likely to have been selected to work at long distances. This considerations might help in explaining the failure of two of the repetitions.

In Responses to the reviewers the author also writes: Not sure which statement the reviewer referred to (page 10 line 17: The attraction of adult male Q-fly to raspberry ketone (RK), emitted by some plants, is related to sexual performance since males exposed to RK can gain substantial mating advantages (Shelly 2010)).

I apologize because I did not exactly report the statment I was referring to. I found various sentences in the paragraph that are speculative.

In particular, the sentence: “Since females can provide their offspring with the genes that are optimal for survival, they are presumably choosing those males for mating who provide the best direct or indirect benefit (“good genes”) for their offspring” is too much speculative. This is not a general rule because females cannot recognize most of the genes of a male and, on the contrary, it is known that certain phenotypes associated to sexual selection are not convenient for the survival of males and progeny. Selection is more generally a consequence of a choice and not a choice.

Later in the same paragraph: “These facts could explain why complete disappearance of the male attraction to CL did not occur in my study” is an hypothesis but, first, the Author has to consider that progeny can inherit the gene variant responsible for the responsiveness to the attractant from the females, that are not selected in this study. You have to consider that in tephritids males are the homogametic sex (ZZ) and females the heterogametic sex (ZW).

These limitations to the selection process maybe deserve to be better higlighted in the discussion section.

Author Response

Dear reviewer,

Thank you for clarification on comments.  

I appreciate the careful review and constructive suggestions.

Following this letter are the reviewer comments with my responses in bold italics, including how and where the text was modified. Changes made in the manuscript are marked using track changes.

Thank you for your time and consideration.

Sincerely,

Maryam Yazdani

“So, we assume unresponsiveness to CL is likely to be a recessive trait” is an hypothesis but there are no data in the article to support it. Therefore, I suggest to remove the sentence from this summary.

“So, we assume unresponsiveness to CL is likely to be a recessive trait” is deleted from the summary and I added my results ” I demonstrated that relaxing artificial selection results in the loss of 35-46% of the selection response after further two generations.”

I Deleted “Potentially productive avenues of future research are discussed in the closing section.” to meet the 100 words limit

In Responses to the reviewers the author writes: “Shelly 1979 used the larger environment for their selection experiments, so I assumed that more controlled environment would give a better result. In my study the proportion of unresponsive Q-fly males male Q-fly non-responders to CL increased to 54%-65% between the second and fifth artificial selection cycles, while for B. dorsalis (Shelly, 1997), in two of the four test lines, the proportion of unresponsive males non-responding males increased from approximately 5% to 25% in 12 generations in the field cages”.

The studies are not comparable because different species and different attractant are involved. Reported results are not sufficient to generalize the mechanism of selection. Maybe, my comment was not clear but, what I meant was that, in a small environment, it is easier for an unresponsive male to land on the attractant even if not, or just less, attracted. Also, small environments have a higher possibility to be saturated by an attractant that is likely to have been selected to work at long distances. This considerations might help in explaining the failure of two of the repetitions.

 Thanks for explanation. When I looked at literatures for designing the experiment, I found that Shelly successfully developed unresponsive lines in two of the four test lines, Shelly didn’t point to any explanation for why it occurred. At that time I assumed that running the selection experiment in the large open arena might cause this variance. That’s why I designed my experiment in smaller arena and under controlled conditions. However, I got similar results, that why I pooled the collected data for all four replicates to find out if the pattern of their response would be different. However, after pooling data from 4 replicates the patter was more pronounced and showed lower variation. Although I agree with the reviewer comment in which there is a higher possibility in the small arena, based on my observation and the results from my filed work I don’t think this would explain why we got different level of response for 2 replicates. Also as I developed low and high responsive lines and the conditions for all tested flies were similar, I don’t think that the size of arena would explain this variation.  

In Responses to the reviewers the author also writes: Not sure which statement the reviewer referred to (page 10 line 17: The attraction of adult male Q-fly to raspberry ketone (RK), emitted by some plants, is related to sexual performance since males exposed to RK can gain substantial mating advantages (Shelly 2010)).

I apologize because I did not exactly report the statment I was referring to. I found various sentences in the paragraph that are speculative.

In particular, the sentence: “Since females can provide their offspring with the genes that are optimal for survival, they are presumably choosing those males for mating who provide the best direct or indirect benefit (“good genes”) for their offspring” is too much speculative. This is not a general rule because females cannot recognize most of the genes of a male and, on the contrary, it is known that certain phenotypes associated to sexual selection are not convenient for the survival of males and progeny. Selection is more generally a consequence of a choice and not a choice.

Later in the same paragraph: “These facts could explain why complete disappearance of the male attraction to CL did not occur in my study” is an hypothesis but, first, the Author has to consider that progeny can inherit the gene variant responsible for the responsiveness to the attractant from the females, that are not selected in this study. You have to consider that in tephritids males are the homogametic sex (ZZ) and females the heterogametic sex (ZW).

These limitations to the selection process maybe deserve to be better higlighted in the discussion section.

I agree, this paragraph is deleted and modified as:

In fact, phytochemical lures are associated with sexual selection governed by both female preference and male competitive mechanisms (Kumaran, Balagawi, Schutze & Clarke,2013). When females were allowed to assess male attractiveness and choose appropriate mates, divergence in the population did not increase (Steiger & Stökl’s, 2014). Also, Hein et al. (2011) observed that a multivariate male trait, preferred by D. serrata females, can respond to selection and result in the maintenance of male mating success (Hein et al., 2011). As Q-fly females are not responsive to CL, therefore they were not subjected to selection in my study. However, the progeny can inherit the gene variant responsible for the responsiveness to the attractant from both sexes. These facts could explain why complete disappearance of the male attraction to CL did not occur in my study.
